# Coarse-grained component concurrency in Earth System modeling: parallelizing atmospheric radiative transfer in the GFDL AM3 model using the Flexible Modeling System coupling framework

V. Balaji[1], Rusty Benson[2], Bruce Wyman[2], and Isaac Held[2]

[1]Princeton University, Cooperative Institute of Climate Science, Princeton NJ, USA
[2]National Oceanic and Atmospheric Administration/Geophysical Fluid Dynamics Laboratory (NOAA/GFDL), Princeton NJ, USA

*Correspondence to:* V. Balaji (`balaji@princeton.edu`)

**Abstract.** Climate models represent a large variety of processes on a variety of time and space scales, a canonical example of multi-physics multi-scale modeling. Current hardware trends, such as Graphical Processing Units (GPUs) and Many-Integrated Core chips (MICs), are based on, at best, marginal increases in clock speed, coupled with vast increases in concurrency,

particularly at the fine grain. Multi-physics codes face particular challenges in achieving fine-grained concurrency, as different physics and dynamics components have different computational profiles, and universal solutions are hard to come by.

We propose here one approach for multi-physics codes. These codes are typically structured as *components* interacting via software frameworks. The component structure of a typical Earth system model consists of a hierarchical and recursive tree of components, each representing a different climate process or dynamical system. This recursive structure generally encompasses

a modest level of concurrency at the highest level (e.g atmosphere and ocean on different processor sets) with serial organization underneath.

We propose to extend concurrency much further by running more and more lower- and higher-level components in parallel with each other. Each component can further be parallelized on the fine grain, potentially offering a major increase in scalability of Earth system models.

We present here first results from this approach, called Coarse-grained Component Concurrency, or CCC. Within the Geophysical Fluid Dynamics Laboratory (GFDL) Flexible Modeling System (FMS), the atmospheric radiative transfer component has been configured to run in parallel with a composite component consisting of every other atmospheric component, including the atmospheric dynamics and all other atmospheric physics components. We will explore the algorithmic challenges involved in such an approach, and present results from such simulations. Plans to achieve even greater levels of coarse-grained

concurrency by extending this approach within other components such as the ocean, will be discussed.

## 1 Introduction

Climate and weather modeling have historically been among the most computationally demanding domains using high-performance computing. Its history parallels that of modern computing itself, starting with experiments on the ENIAC (Platzman, 1979) and continuing through several changes in supercomputing architecture, including the vector and parallel eras.

The transition from vector to parallel computing was "disruptive", to use a currently popular term. The computing industry itself was transitioning from being primarily responsive to military and scientific needs, to being dominated by a mass market demanding cheap and ubiquitous access to computing. This gradually led to demise of specialized computational machines and the high end of the market also being dominated by clusters built out of mass-market commodity parts (Ridge et al., 1997; Sterling, 2002).

The community weathered the challenge well, without significant loss of pace of scientific advance. More narrowly stated, Earth System Models (ESMs) continued to demonstrate continual increases in both resolution and complexity across the vector/parallel transition. For example, the typical resolution of climate models used for the Intergovernmental Panel on Climate Change (IPCC) assessments and their complexity (the number of feedbacks and phenomena simulated), exhibits a steady increase from the 1990 First Assessment Report (known as FAR) to the Fourth Assessment Report (AR4) (Solomon, 2007, see e.g the iconic *Figure 1.2*[1] and *Figure 1.4*[2] from the Summary for Policymakers).

A second disruption is upon us in the current era. Current computing technologies are based on increased concurrency of arithmetic and logic, while the speed of computation and memory access itself has stalled. This is driven by many technological constraints, not least of which is the energy budget of computing (Cumming et al., 2014; Charles et al., 2015; Kogge et al., 2008). These massive increases in concurrency pose challenges for high-performance computing (HPC) applications: an era where existing applications would run faster with little or no effort, simply by upgrading to newer hardware, has ended. Substantial recoding and re-architecture of applications is needed. This poses particular challenges to applications such as climate modeling, where we must simulate many interacting subsystems. The state of play of climate computing in the face of these challenges, is surveyed in Balaji (2015) and references therein: for the current generation of technology, the gains to be had seem modest, and the effort of recoding immense. Whether we will continue to demonstrate continued increases in resolution and complexity through this transition remains to be seen.

ESMs[3] are canonical multi-physics codes, with many interacting components, each often built by independent teams of specialists. The coupling of these components, while respecting algorithmic constraints on conservation and numerical accuracy, is a scientific and technological challenge unto itself. Within each component of an ESM, code is parallelized using multiple techniques, including distributed and shared memory parallelism, as well as vector constructs.

---

[1] https://www.ipcc.ch/publications_and_data/ar4/wg1/en/figure-1-2.html

[2] https://www.ipcc.ch/publications_and_data/ar4/wg1/en/figure-1-4.html

[3] Note that we are using the term "ESM" generically to denote any level in the hierarchy of complexity of weather and climate models: ranging from single-component models, e.g an atmospheric general circulation model, to models that include coupling with the ocean and land, biospheres, an interactive carbon cycle, and so on. See Figure 2.

Multi-physics codes are particularly challenged by the coming trends in HPC architecture. These codes typically involve many physical-chemical-biological variables (complexity) and associated process representations in code. Computational load is evenly distributed across many components, each embodying different physics: there are no performance "hotspots". This also means that fresh operators and operands – embodied in physics subroutines and associated variables – are constantly being transferred to and from memory with each context switch, and locality and reuse are hard to achieve. This is particularly unsuited to the novel architectures currently on the horizon. These include graphical processing units (GPUs) which can concurrently process $\mathcal{O}(100)$ data streams following the same instructions sequence, and the Many-Integrated Core (MIC) architecture, which allows many ($\mathcal{O}(100)$) execution threads to access the same memory. These hardware trends have made the cost of data movement prohibitive relative to computing itself, thus strongly favoring codes where both instructions and data have a high rate of reuse and computational intensity (ratio of floating-point operations to memory operations). Algorithms that exhibit *fine-grained concurrency*, where multiple computationally intensive and concurrent data streams follow the same instruction sequence, are best adapted to the emerging architectures of this decade.

The next computational landmark at this juncture is the "exascale", $\mathcal{O}(10^{18})$ operations per second. Given the earlier discussion on the stalling of Moore's Law, the rate of atomic arithmetic operations is still $\mathcal{O}(10^9)$ per second, thus requiring us to achieve $\mathcal{O}(10^9)$ concurrency. While continuing to extend physical and dynamical algorithms toward the fine-grained concurrency of the coming era, we believe multi-physics codes must also attempt to share the available concurrency across many physical components, in order to best exploit these new systems. Of the many factors of 10 increase in performance needed to get to the exascale, we believe at least one can come from component organization. We propose here a major architectural change in the construction of coupled models, such as ESMs. We demonstrate here a possible approach to extending the current rather modest amount of concurrency among ESM components (typically 2-4 top-level realms such as atmosphere, ocean, and land) to a more massive increase in *coarse-grained component concurrency (CCC)*.

In this study, we examine the radiation component (which computes radiative transfer in the atmosphere in response to dynamically evolving concentrations of radiatively active chemical species) of the GFDL Flexible Modeling System (FMS). This is a relatively expensive component of the atmospheric physics, and is usually run at a much coarser timestep than the rest of the atmosphere (temporal subsampling), purely for expediency rather than any physical justification. Other approaches to reducing the computational burden of radiative transfer include subsampling in the spectral domain (Pincus and Stevens, 2009; Bozzo et al., 2014) or in the spatial as well as the temporal domain (Morcrette, 2000; Morcrette et al., 2008). Some of these methods have been shown to be effective over short timescales (e.g numerical weather prediction and medium-range forecasting) but contribute to model bias over climate timescales. Adaptive methods that tune the subsampling by examining the degree of spatial and temporal correlation in model fields have also been proposed (Manners et al., 2009).

We focus here on temporal subsampling. This purely expedient choice of timestep has been shown by Pauluis and Emanuel (2004) to be a potential source of instability and bias in radiating atmospheres. Xu and Randall (1995) have also shown that this problem gets considerably worse as the resolution of models increases. A useful way to think about this is that using different timesteps for the radiation component vis-à-vis the rest of the physics creates a discrepancy between the cloud field and the "cloud shadow field" seen by the radiation component, which can lead to numerical issues. Our method permits us to reduce

the timestep to match the rest of the atmosphere, *with the same time to solution*, at a modest computational cost in terms of allocated processors. This method does not rule out subsampling along other dimensions (spatial or spectral), which may be superimposed as well in future developments. The effects of subsampling are not fully understood yet, and further study is needed to understand how results converge as various forms of subsampling are eliminated. That said, subsampling is clearly a matter of expediency and reducing computational expense: there is no case at all to be made that it is in any way numerically or physically superior to the alternative.

The structure of the paper is as follows. In Section 2 we briefly review current approaches to parallelism in ESMs, particularly in the coupling framework. In Section 3 we describe our approach to coarse-grained concurrency, how it is achieved without increasing data movement. In Section 4 we show results from standard AMIP (the Atmospheric Model Intercomparison Project: Gates, 1992) simulations using the CCC approach. The associated computational results show decreased time to solution for concurrent versus serial approaches in otherwise identical physical formulations, and the ability to run with a much smaller radiation timestep without increasing the time to solution. Finally, in Section 5 we discuss plans and prospects for extending this approach further within FMS, and its potential application on novel architectures.

## 2   Concurrency in Earth System Models

Weather and climate modeling have always been in the innovative vanguard of computing, dating all the way back to the origins of modern computing in John von Neumann's pioneering studies (Dahan-Dalmedico, 2001). The ability to apply instruction sequences to multiple data streams – concurrency – has long been a cornerstone of performance engineering. The pioneering vector processors of Seymour Cray's era in the late 1970s allowed a data stream to flow through a hardware innovation known as *vector registers*, which allowed the same instruction sequences to apply to each succeeding element in the data stream, known as SIMD (single-instruction multiple-data). Over time, vectors grew to support extremely complex programming sequences, evolving into single-program, multiple-data, or SPMD.

In the 1980s, machines such as the Cray X-MP (the MP stood for multi-processor) were introduced. Here for the first time *parallelism* appears at a very high-level, allowing the concurrent execution of multiple tasks, which were themselves SPMD vector programs. This was the first introduction of coarse-grained concurrency. This led to the development of the MPMD framework: multiple-program, multiple-data. Soon after, distributed computing, consisting of networked clusters of commodity computers, known as symmetric multi-processors (SMPs), began to dominate HPC, owing to the sheer advantage of the volume of the mass market.

To take advantage of distributed computing, new techniques of concurrency began to be developed, such as domain decomposition. Here the globally discretized representation of physical space in a model component is divided into domains and assigned to different processors. Data dependencies between domains are resolved through underlying communication protocols, of which the Message-Passing Interface MPI (Gropp et al., 1998) has become the *de facto* standard. The details of message-passing are often buried inside software frameworks (of which the GFDL Flexible Modeling System, described in Balaji (2012) is an early example), and this convenience led to the rapid adoption of distributed computing across a wide va-

riety of applications. Within the distributed domains, further fine-grained concurrency is achieved between processors sharing physical memory, with execution threads accessing the same memory locations, using protocols such as OpenMP (Chandra et al., 2001).

Climate computing has achieved widespread success in the distributed computing era. Most ESMs in the world today are MPMD applications using a hybrid MPI-OpenMP programming model. At the highest end, ESMs (or at least, individual components within them, see e.g Dennis et al., 2012, S.-J. Lin and C. Kerr, private communication) have been run on $\mathcal{O}(10^5)$ distributed processors and $\mathcal{O}(10)$ shared-memory execution threads, which places them among the most successful HPC applications in the world today (Balaji, 2015). Even higher counts are reported on some leadership machines, but these are more demonstrations than production runs for science (e.g Xue et al., 2014).

## 2.1 Coupling algorithms in Earth System Models

There are diverse component architectures across Earth System Models (Alexander and Easterbrook, 2015), but they nonetheless share common features for the purposes of discussion of the coupling algorithms. Consider the simplest case, that of two components, called $A$ and $O$ (symbolizing atmosphere and ocean). Each has a dependency on the other at the boundary. When the components execute serially, the call sequence can be schematically represented as:

$$A^{t+1} = A^t + f(A^t, O^t) \tag{1}$$
$$O^{t+1} = O^t + g(A^{t+1}, O^t) \tag{2}$$

where $f()$ and $g()$ nominally represent the feedbacks from the other component, and the superscript represents a discrete timestep. Note that in the second step, $O$ is able to access the updated state at $A^{t+1}$. This is thus formally equivalent to Euler forward-backward time integration, or Matsuno timestepping, as described in standard textbooks on numerical methods (e.g Durran, 1999).

In a parallel computing framework, now assume the components are executing concurrently. (Figure 1 shows the comparison of serial and concurrent methods in parallel execution.) In this case, $O$ only has access to the *lagged* state $A^t$.

$$A^{t+1} = A^t + f(A^t, O^t) \tag{3}$$
$$O^{t+1} = O^t + g(A^t, O^t) \tag{4}$$

Thus, the results will not be identical to the serial case. Furthermore, while we cannot undertake a formal stability analysis without knowing the forms of $f$ and $g$, this coupling algorithm is akin to the Euler forward method, which unlike Matsuno's method is formally *unconditionally unstable*. Nevertheless, this parallel coupling sequence is widely, perhaps universally, used in today's atmosphere-ocean general circulation models (AOGCMs). This is because for the particular application used here,

that of modeling weather and climate, we find that the system as a whole has many physical sources of stability.[4] Radiative processes are themselves a source of damping of thermal instability, and we also note that within each component there are internal processes and feedbacks which are often computed using implicit methods, and other methods aimed at reducing instability. This is nonetheless a reason for caution, and in Section 5 we will revisit this issue in the context of future work.

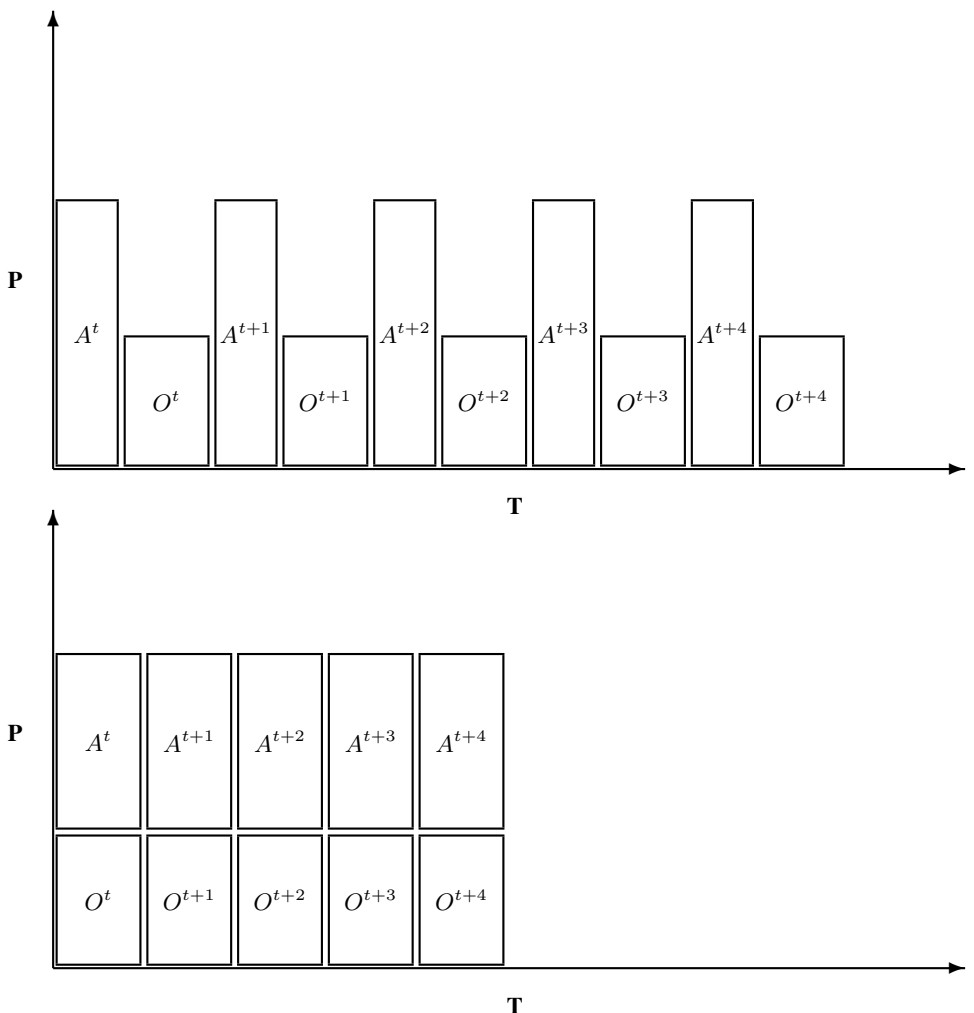

**Figure 1.** Serial and concurrent coupling sequences, with time on the X-axis and processors on the Y-axis. In the serial case, both components may not scale to the same processor count, leaving some processors idle. Note that in the concurrent coupling sequence below, $O^{t+1}$ only has access to the lagged state $A^t$.

---

[4] We are familiar with things that work in theory, but not in practice... this is something that works in practice but not in theory! This is a good example of the opportunistic nature of performance engineering.

This discussion has introduced the notions underlying serial and concurrent coupling in the context of two components $A$ and $O$. An actual ESM has many other components, such as land and sea ice. Components themselves are hierarchically organized. An atmosphere model can be organized into a "dynamics" (solutions of fluid flow at the resolved scale) and "physics" components (subgrid scale flow, and other thermodynamic and physical-chemical processes, including those associated with clouds and subgridscale convection, and the planetary boundary layer). Similarly the land component can be divided into a hydrology and a biosphere component, and the ocean into dynamics, radiative transfer, biogeochemistry, and marine ecosystems. A notional architecture of an ESM is shown in Figure 2. Different ESMs around the world embody these differently in code; this figure is not intended to define the software structure of all ESMs, which tend to be quite diverse (Alexander and Easterbrook, 2015).

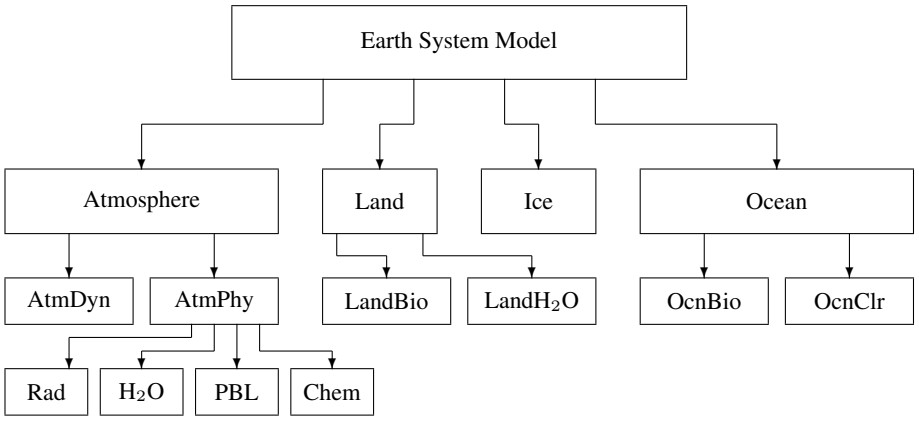

**Figure 2.** Notional architecture of an Earth System Model, with components embodying different aspects of the climate system, hierarchically organized. Models on a hierarchy of complexity ranging from single-component (e.g atmosphere-only) models to full-scale coupled models with an interactive biosphere, are often constructed out of a palette of components within a single modeling system.

How the notional architecture of Figure 2 gets translated into a parallel coupled ESM code is quite problem-specific. As the science evolves and computing power grows, the boundary of what is resolved and unresolved changes. Also, models grow in sophistication in terms of the number of processes and feedbacks that are included.

For the purposes of this study, we describe the actual code architecture of the GFDL Flexible Modeling System (FMS). The atmosphere and ocean components are set up to run in parallel in distributed memory, communicating on the slow coupling timestep $\Delta t_{\mathrm{cpld}}$, on the order of $\Delta t_{\mathrm{cpld}} = 3600$ sec for its flagship application, decadal-centennial climate change. Within the slow coupling loop, the atmosphere communicates on a fast coupling timestep $\Delta t_{\mathrm{atm}}$ with a typical value of 1200 sec, set by the constraints of atmospheric numerical and physical stability.

As the land and ocean surfaces have small heat capacity, reacting essentially instantaneously to changes in atmospheric weather, stability requires an implicit coupling cycle. The implicit coupling algorithm requires an down-up sweep through the

atmosphere and planetary (land and ocean) surface systems, for reasons detailed in Balaji et al. (2006). The parallel coupling architecture of FMS is shown in Figure 3.

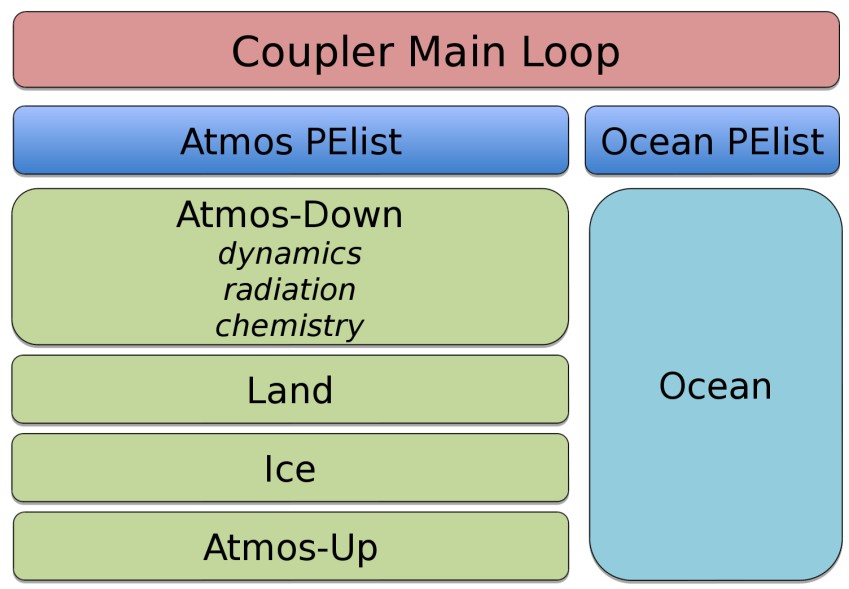

**Figure 3.** FMS parallel coupling architecture in processor-time space, with processors across, and time increasing downward. Components have different horizontal and vertical extents to indicate the degree of parallelism and time of execution, though these extents are notional and not to be interpreted as drawn to scale. Within a single executable for the entire coupled system, the atmosphere and ocean components run concurrently in distributed memory (MPI, indicated in deep blue). Within the atmosphere stack, components execute serially, including tight coupling to the land and ice-ocean surface. The down-up sequence of implicit coupling is explained in Balaji et al. (2006). These components internally shared-memory (OpenMP) coupling, indicated in light green. The ocean component at the present time is MPI-only, indicated in light blue.

The "Atmos-Up" step is quite lightweight, including adjustments to the atmospheric state imposed by moist physics, and completing the up-sweep of a tridiagonal solver for implicit coupling of temperature and other tracers, as described in Balaji et al. (2006). The bulk of the atmospheric physics computational load resides in the "Atmos-Down" step.

The atmospheric radiation component is a particularly expensive component of atmospheric physics, which is why it was chosen as the target for increasing coupling concurrency in FMS. This component is described below.

## 2.2 The radiation component in FMS

The radiation component in FMS is one of the most expensive components within the atmospheric physics. It consists of short-wave and longwave radiation components. The shortwave component is based on Freidenreich and Ramaswamy (1999), where "line-by-line" (LBL) radiative transfer calculations have been grouped into pseudo-monochromatic bands, and shown in bench-

mark calculations to provide a similar response to the benchmark LBL results. The calculations have a strong dependency on the evolving (through advection, cloud processes, and chemistry) state of radiatively active species in the atmosphere, including atmospheric water vapor, $CO_2$, $O_3$, aerosols, and condensed water fields, in addition to the basic physical state variables. The longwave radiation components (Schwarzkopf and Ramaswamy, 1999) similarly use approximations for computational effi-
ciency, and also interact strongly with atmospheric liquid- and gas-phase chemical species, including water vapor and clouds.

 The species shared between atmospheric physics, chemistry, and radiation are referred to as *tracers*, a term applied to 3D model fields (in atmosphere or ocean) that are advected by the evolving dynamics, and participating in physics and chemistry processes at individual gridpoints.

Despite the simplifying approximations, the radiation component remains prohibitively expensive. As a result, this compo-
nent is stepped forward at a slower rate than the rest of the atmospheric physics, with $\Delta t_{\mathrm{rad}} = 10800$ sec, or $9 * \Delta t_{\mathrm{atm}}$, as a typical value. The planetary surface albedo, whose time evolution is a function of solar zenith angle only, alone is stepped forward on the atmospheric timestep $\Delta t_{\mathrm{atm}}$. This means that at intermediate (non-radiation) atmospheric timesteps, the radiation is responding to a lagged state of atmospheric tracers, which may be as much as $\Delta t_{\mathrm{rad}} - \Delta t_{\mathrm{atm}}$ ($\sim$3 h) behind.

This timestep discrepancy is vexing, but most climate models around the world make a similar compromise, with a radiation
timestep longer than the physics timestep. If the promise of massive increases in concurrency on future architectures is kept, a concurrent radiation component may offer a way forward. Simultaneously, we may be able to decrease the discrepancy between $\Delta t_{\mathrm{rad}}$ and $\Delta t_{\mathrm{atm}}$, and bring us toward more physical consistency between the radiative and physico-chemical atmospheric states (Pauluis and Emanuel, 2004; Xu and Randall, 1995).

## 3    Coarse-grained component concurrency

Before we begin describing a method for casting the radiation code in FMS as a concurrent component, we need to describe the current methodology shown in Figure 3. Concurrency between atmosphere and ocean component on the slow coupling timestep is achieved using distributed computing techniques, with the components running on separate processor sets or *PElists*. In FMS terminology, a PE or *processing element* is a unit of hardware supporting a single execution thread, sometimes called a *core*. A PElist is synonymous with a "communicator" in MPI terminology, and lists the PEs assigned in distributed memory processing.
Each PE in a PElist can spawn multiple shared-memory execution threads. These threads are assigned to other PEs to avoid contention. Coupling fields are transferred between atmosphere and ocean using the exchange grid (Balaji et al., 2006) and message passing. Within the atmosphere component, shared-memory parallelism using OpenMP is already implemented. For the dynamics phase, the OpenMP acts on individual loops, some of which may contain calls to subroutines or comprised of large programmatic constructs. These include regions where concurrency is on slabs (horizontally tightly-coupled) and others
organized in columns (vertically tightly coupled).

Unlike the dynamics, the physics is organized entirely columnwise, and individual columns – the $k$ index in an $(i, j, k)$ discretization – have no cross-dependency in $(i, j)$ and can execute on concurrent fine-grained threads. The arrays here can be organized into groups of vertical columns, or *blocks*, that can be scheduled onto the same OpenMP threads at a high (coarse)

level – meaning a single thread will persistently see the block (thus assuring thread data affinity) through the complete Atmos-Down phase (*sans* dynamics), and again through the Atmos-Up phase.

We now come to the reorganization of components for CCC in the current study. Understanding that the radiation is but one phase of the physics, which is already utilizing blocks and OpenMP threads, it makes sense to extend the concept and have the radiation run concurrently in a separate group of OpenMP threads. In the concurrent radiation architecture, shown in Figure 4, the decision was made to utilize *nested* OpenMP, instead of a flat threadpool. Each MPI-rank assigned to the atmosphere PElist starts up an OpenMP region with two threads, one to act as the master for the radiation and the other to drive the non-radiation components. These "master" threads are able to utilize the nested OpenMP constructs, to start up a number of atmosphere threads ($A$) and radiation threads ($R$) where the total numbers of threads $T = A + R$. For a given value of $T$, $A$ and $R$ can be dynamically adjusted during the course of a run to achieve optimal load-balance. Because the radiation and atmosphere concurrency occur at a high level, the memory space is unique to each component and furthermore, the memory space is unique to each block. This separation of memory spaces ensures there are no performance deficiencies due to cache coherency effects (false sharing or cache invalidations). A single data synchronisation point (copy) at the end of each atmospheric time step ensures that the atmospheric and radiation components remain completely independent.

In the limit, we could create blocks containing a single column, so that $A$ and $R$ both equal the number of columns in the domain, and $T = 2A$. But the overheads associated with moving in and out of threaded regions of code must be amortized by having enough work per OpenMP thread instance. Current processors rely on having a moderate number of data elements to achieve best performance by hiding the various latencies for each instruction, including time spent waiting for operands to be loaded to memory registers. Empirically, on current technology, we have found optimal results using blocks of $\mathcal{O}(32)$ columns.

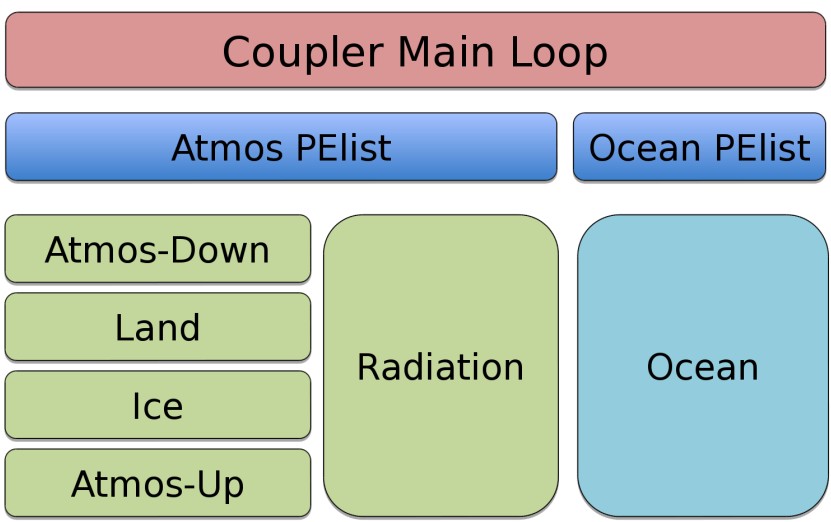

**Figure 4.** Concurrent radiation architecture. See Figure 3 for comparison, and explanation of legends.

## 4 Results

### 4.1 Results from AMIP runs

The model utilized here is based on AM3, the atmosphere-land component of the GFDL Climate Model Version 3 (CM3) model
(Donner et al., 2011), a model with a relatively well-resolved stratosphere, with a horizontal resolution of approximately
100 km and 48 vertical levels. Here the original AM3 has been modified to include an experimental cumulus convection
scheme, and a reduced chemistry representation including gas and aqueous-phase sulfate chemistry from prescribed emissions
(Zhao et al., 2016). This model is forced using observed sea surface temperatures (SSTs) as a lower boundary condition over a
20-year period 1981-2000. The three experiments described here are:

1. the control run (CONTROL) using serial radiation with a radiative time step $\Delta t_{\mathrm{rad}}$ of 3 hours ($\Delta t_{\mathrm{rad}} = 9\Delta t_{\mathrm{atm}}$ where
   $\Delta t_{\mathrm{atm}}$ = 1200 s is the time scale on which the atmospheric state is updated;

2. serial radiation (SERIAL) using $\Delta t_{\mathrm{rad}} = \Delta t_{\mathrm{atm}}$ = 1200 s; and

3. concurrent radiation (CONCUR) also using $\Delta t_{\mathrm{rad}} = \Delta t_{\mathrm{atm}}$ = 1200 s. The difference between the SERIAL and CON-
   CUR experiments shows the impact of concurrent coupling (the radiation sees the lagged atmospheric state), while
   the CONTROL and SERIAL experiments only differ in the radiative time step. We could of course attempt CONCUR
   while maintaining $\Delta t_{\mathrm{rad}}/\Delta t_{\mathrm{atm}} = 9$, but because of the lagged timestep, this is not recommended: the atmospheric and
   radiative states would be effectively 21600 s, or 6 h, out of synchrony.

All versions of the model utilize what we refer to as a solar interpolator. At the beginning of a radiative time step, the
distribution of radiatively-active atmospheric constituents, such as water vapor and clouds, are input into computations of
both shortwave and longwave radiative fluxes. When $\Delta t_{\mathrm{rad}}$ is longer than $\Delta t_{\mathrm{atm}}$ all solar fluxes are rescaled every $\Delta t_{\mathrm{atm}}$ by
normalizing by the incident solar radiation using the zenith angle appropriate for that atmospheric time step. Any sensitivity
to the $\Delta t_{\mathrm{rad}}$ radiation time step is due to the fact that the radiatively-active constituents are held fixed for the duration of that
time step and not due to neglected changes in the incoming solar flux (Morcrette, 2000).

We show here the effects of changing the radiation timestep on precipitation and top-of-atmosphere radiative fluxes in the
AMIP simulation. Figure 5 shows the annual mean precipitation bias for the three experiments with respect to the Global
Precipitation Climatology Project (GPCP) v2.2 (Adler et al., 2003) climatology. Panel (b) shows the difference between
CONTROL and observational precipitation climatology, while (c) and (d) show the model-model differences between the
CONTROL run and the SERIAL and CONCUR climatologies. The difference in the pattern of annual mean precipitation
due to the change in time step (Figure 5, Panels c and d) is negligible compared to the difference between the model and
observations. Panels (c) and (d) are similar to first approximation, indicating that the difference in precipitation pattern due

to the choice of serial versus concurrent integration is smaller than the difference due to the radiation time step and even less significant as compared to the difference between model and observations.

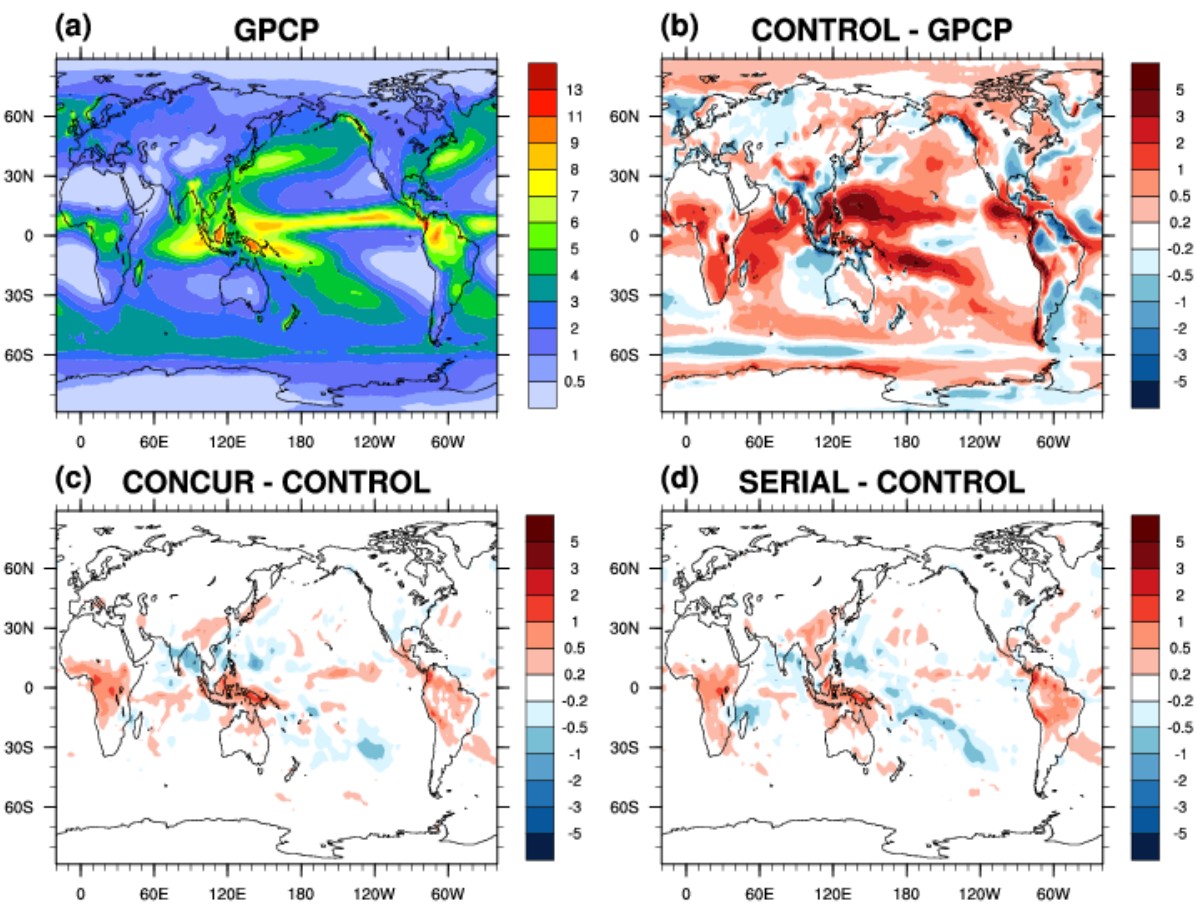

**Figure 5.** Comparison of model climatologies against GPCP precipitation. (a) shows the GPCP climatology, and (b) the model climatological biases for the CONTROL run. Panels (c) and (d) show model-model difference versus CONTROL for CONCUR and SERIAL runs respectively, plotted on the same color scale as (b).

The more significant difference in the simulation due to the change in radiation time step is in the energy balance at the top of the atmosphere between absorbed shortwave and outgoing longwave radiation. The simulated annual mean pattern of this net energy flux is displayed in Figure 6 in analogous format to Figure 5. The energy flux data is CERES EBAF edition 2.8 satellite data (Loeb et al., 2009). Neither the radiation time step nor the choice of serial versus concurrent integration modifies the geographical pattern of model bias significantly, but they do alter the simulation with a fairly spatially uniform

offset. Again the details of all the experiments are similar but a closer examination of the global mean biases show the SERIAL and CONCUR cases (Figure 6, Panels c and d) differ from the CONTROL by about +3.1 to +3.6 W/m$^2$. This magnitude of flux difference would have a significant effect on a coupled atmosphere-ocean model. (Compare the change in flux due to a doubling of $CO_2$ concentrations, holding radiatively active atmospheric constituents fixed, of about 3.5 W/m$^2$). Nearly all of this difference is in the absorbed shortwave, most of which occurs over the oceans and tropical land areas. The source of this difference is primarily clouds and to a lesser extent water vapor, as determined by examining the clear-sky energy balance, a model diagnostic. The diurnal cycle of clouds and solar radiation appear to be the key factors in determining the sign and size of these responses. The diurnal peak in clouds over the oceans typically occurs close to sunrise, so there is a downward trend in cloudiness on average at the peak in incoming solar radiation. Therefore the CONTROL case sees more cloudiness over the longer radiative time step, therefore leading to more reflection by clouds and less absorption of shortwave radiation at the surface and in the atmosphere.

To be a viable climate model, the global mean top of atmosphere energy balance has to be small, less than 1 W/m$^2$, to avoid unrealistic climate drift. This global energy balance is tuned in all models, as our ability to simulate the Earth's cloud field from first principles is inadequate to generate a radiation field with the required fidelity. Parameters in the model that control the cloud simulation and that are not strongly constrained by observations are used in this tuning process (see e.g Hourdin et al., 2016, for a description of the tuning process). The model simulations displayed here have not been "retuned" so as to isolate the effects of the radiation time step and coupling strategy. In our experience a retuning of 3-4 W/m$^2$ is viable, but large enough that the resulting changes in other aspects of the model simulation can be non-negligible, emphasizing the importance of algorithms that make the reduction in the radiative time step less onerous computationally. However, the difference between serial and concurrent coupling of 0.5 W/m$^2$ is well within the range in which retuning has marginal impact on other aspects of the simulation, encouraging examination of the performance of the concurrent option.

## 4.2 Scaling and performance results

Comparisons of the computational performance of the 3 configurations (CONTROL, SERIAL and CONCUR) were performed on the NOAA supercomputer Gaea. Recall that CONTROL is intrinsically computationally less expensive as the $\Delta t_{\rm rad} = 9 * \Delta t_{\rm atm}$ setting implies that the radiation code is executed very seldom. As the results of Section 4.1 suggest that we should shorten $\Delta t_{\rm rad}$ if we can, the aim is now to recover the overall model throughput (measured in simulated years per day, or SYPD) of CONTROL using the CONCUR configuration with the shorter timestep $\Delta t_{\rm rad} = \Delta t_{\rm atm}$, but at a higher processor count. The other measure in our comparison is that of the integrated processor-time computational resource request, measured in compute-hours per simulated year (CHSY). These are key measures of computational cost (time to solution, and resource consumption) used at modeling centers around the world (Balaji et al., 2016).

Initial studies were performed on a machine configuration that used AMD Interlagos processors on Cray's Gemini high-speed interconnect. For any given PE count, we attempt different partitions between processors and threads (MPI and OpenMP) to arrive at the optimal processor/thread layout for that PE count. As Table 1 shows, CONTROL achieved 9.25 SYPD on 1728 PEs. The SERIAL configuration shows the relative cost of radiation to the rest of the model, as shortening $\Delta t_{\rm rad}$ from 10800 s

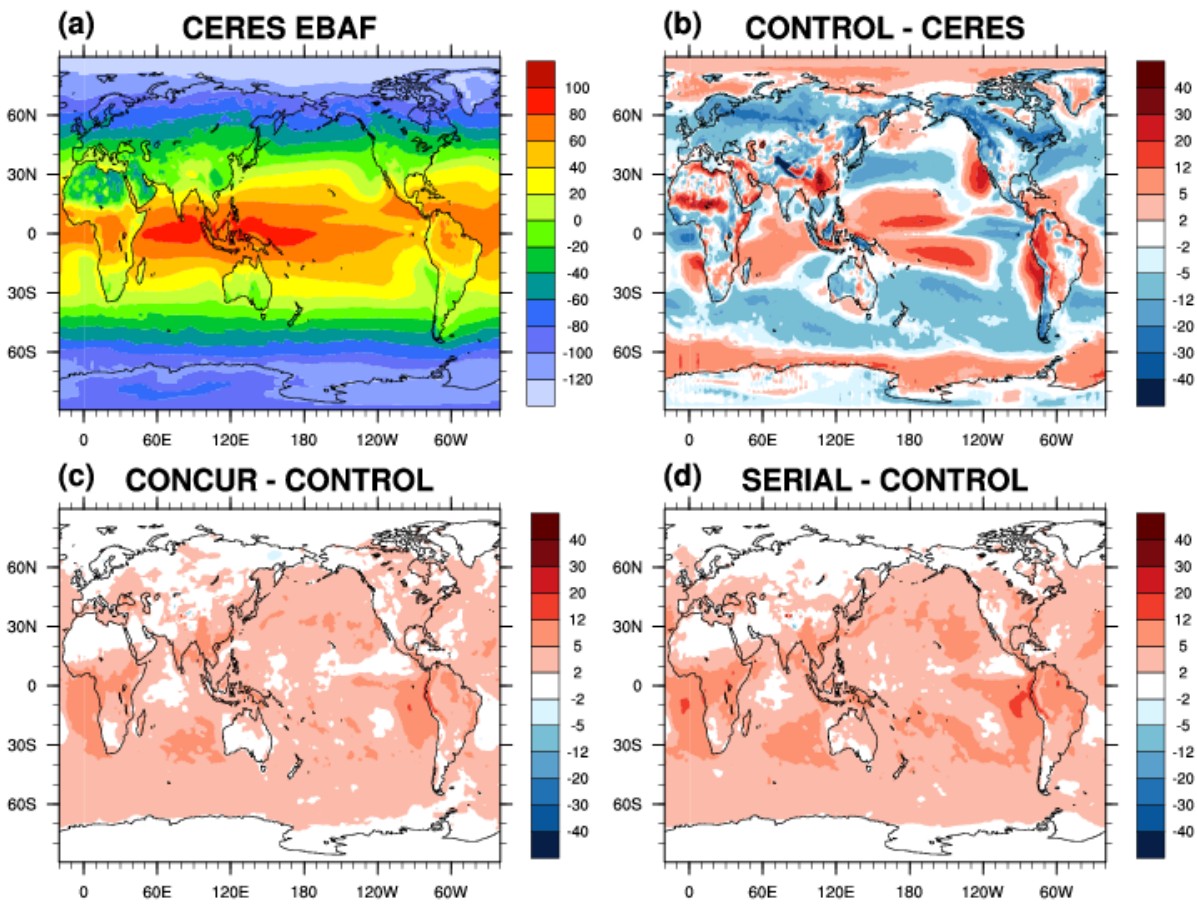

**Figure 6.** Comparison of model climatologies against CERES EBAF v2.8 climatological top-of-atmosphere radiation budget shown in (a). Panel (b) shows model climatological biases for the CONTROL run. Panels (c) and (d) show model-model difference versus CONTROL for CONCUR and SERIAL runs respectively, plotted on the same color scale as (b).

to 1200 s, without changing the processor count, substantially raises the total cost of radiation computations within the code, bringing time to solution down to 5.28 SYPD. Running CONCUR on the same processor count increases this time to 5.9 SYPD. Increasing the processor count to 2592 brings us back to 9.1 SYPD. Thus, one can achieve the goal of setting $\Delta t_{\mathrm{rad}} = \Delta t_{\mathrm{atm}}$ without loss in time to solution (SYPD), at a cost of 1.52X (the CHSY ratio of the two configurations) in resources. Thus
5   decreasing the radiation timestep 9-fold has raised the computational cost by about 50%, indicating that the original cost of radiation in units of CHSY was about 5%. Models where this fraction is higher will derive an even more substantial benefit from the CCC approach. We believe that as computing architectures express more and more concurrency while clock speeds

stall (see Section 5 below), this will be a key enabling technology for maintaining time to solution while increasing parallelism. While these results are for an atmosphere-only model, they can be readily extended to other components in a more complex model. As noted below in Section 5, we are planning to extend the CCC approach to other components including atmospheric chemistry and ocean biogeochemistry.

| Configuration | $\Delta t_{\mathrm{rad}}/\Delta t_{\mathrm{atm}}$ | MPI*OMP | NPES | SYPD | CHSY |
|---|---|---|---|---|---|
| CONTROL | 9 | 864*2 | 1728 | 9.25 | 4483 |
| SERIAL | 1 | 864*2 | 1728 | 5.28 | 7854 |
| CONCUR | 1 | 432*4 | 1728 | 5.90 | 7029 |
| CONCUR | 1 | 648*4 | 2592 | 9.10 | 6836 |

**Table 1.** Performance results from the various configurations discussed. MPI*OMP shows the assignment of MPI processes and OpenMP threads. In the CONCUR cases 2 threads each are assigned to atmosphere and radiation components. NPES is the total PE count (MPI*OMP). SYPD measures throughput in simulated years per day, and CHSY is the computation cost in processor-hours per simulated year (NPES*24/SYPD).

## 5   Summary and Conclusions

We are at a critical juncture in the evolution of high-performance computing, another "disruptive" moment. The era of decreasing time to solution at a fixed problem size, with little no effort, is coming to an end. This is due to the ending of the conventional meaning of Moore's Law (Chien and Karamcheti, 2013), and a future where hardware arithmetic and logic speeds stall, and further increases in computing capacity are in the form of increased concurrency. This comes in the form of heterogeneous computing architectures, where co-processors or accelerators such as Graphical Processing Units (GPUs) provide SIMD concurrency; the other prevalent approach is in the Many-Integrated Core (MIC) architectures, with a vastly increased thread space, an order of magnitude higher than the $\mathcal{O}(10)$ thread parallelism achieved with today's hybrid MPI-OpenMP codes.

It is very likely that radical reimagining of ESM codes will be necessary for the coming novel architectures (Balaji, 2015). That survey of the current "state of play" in climate computing notes that multi-physics codes, which are fundamentally MPMD in nature, are particularly unsuited to these novel architectures. While individual components show some speedup, whole MPMD programs show only modest increases in performance (see e.g Govett et al., 2014; Iacono et al., 2014; Fuhrer et al., 2014; Ford et al., 2014). Other approaches, such as the use of "inexact" computing (Korkmaz et al., 2006; Düben et al., 2014), are still in very early stages.

We have demonstrated a promising new approach for novel and heterogeneous architectures for MPMD codes such as Earth System models. It takes advantage of the component architecture of ESMs. While concurrency has been achieved at the very highest level of ESM architecture shown in Figure 2, the components are themselves MPMD within a hierarchical component architecture.

In the light of our discussion we propose a precise definition of a component as a unit of concurrency. For the purposes of the coarse-grained concurrency (CCC) approach, a *component* may be defined as *one of many units in a multi-physics model, that is itself SIMD for the most part*. While the word has been loosely used earlier, this study has provided guidance on how we should think about components, and thus, this definition will be followed for the rest of the discussion in this section.

Fine-grained parallelism approaches, such as those surveyed in Mittal and Vetter (2015), may be applied within a component as so defined, but are likely to fail above that level. A substantial increase in overall scalability of an ESM may be achieved if several components are run concurrently. We are currently exploring CCC in several other computationally burdensome model components, including atmospheric chemistry and ocean biogeochemistry. It is clear however that this is not an universal solution: given the constraint that concurrent components can only "see" each others' time-lagged state, some components are too tightly coupled to be amenable to the CCC approach.

Furthermore, we have demonstrated a method where multiple components that share a large number of model fields can be run concurrently *in shared memory*. This avoids the necessity of message passing between components that need to synchronize on fine timescales.

We believe the CCC approach will afford very tangible benefits on heterogeneous architectures such as GPUs, and architectures with a wide ($\mathcal{O}$(100-1000)) thread space, such as MICs. In particular:

– Threading within a SIMD component has not been shown to scale beyond a rather modest thread count. By running multiple components within a single thread space, the thread count can be considerably increased.

– Even with additional work in improving the SIMD performance of components, it is clear that some components are better suited to SIMD architectures than others. In a heterogeneous system, with different hardware units, this method may permit different components to be scheduled on the hardware unit to which they are best suited. For example, we could imagine some embarrassingly parallel components executing on a GPU while another, less suited to that architecture, executes on its host CPU (central processing unit).

There remain caveats to this approach. As shown in the discussion of Equation 3 above, the coupling of concurrent components might be formally unstable. We are exploring more advanced time-coupling algorithms, including 3-time-level schemes such as Adams-Bashforth (see Durran, 1999). Such algorithms have been successfully used within the atmospheric dynamical core of FMS, for two-way nesting. In this approach, the coarse- and fine-mesh components execute concurrently rather than serially as in conventional nesting approaches (Harris and Lin, 2013). We are also exploring a combination of the 3-time-level schemes with time-staggering of components, which no longer suffers from formal instability.

.We conclude that coarse-grained concurrency remains a very promising road to the future of Earth System modeling on novel, massively-concurrent HPC architectures.

## 6 Source code and data availability

Source code and data, including model output and performance data, associated with this study are freely available upon request.

*Acknowledgements.* The authors thank Alexandra Jones and Larry Horowitz of NOAA/GFDL for close reading and incisive comments that have greatly improved the quality of the manuscript.

V. Balaji is supported by the Cooperative Institute for Climate Science, Princeton University, under Award NA08OAR4320752 from the National Oceanic and Atmospheric Administration, U.S. Department of Commerce. The statements, findings, conclusions, and recommendations are those of the authors and do not necessarily reflect the views of Princeton University, the National Oceanic and Atmospheric Administration, or the U.S. Department of Commerce. He is grateful to the Institut Pierre et Simon Laplace (LABEX-LIPSL) for support in 2015 during which drafts of this paper were written.

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

If the objection is to the use of the Mercator projection: we agree that it introduces distortions, particularly areal distortion as you approach the poles. While this is known, this projection is customary and in very widespread use in the literature. We do not believe the use in this instance to be egregious, as the results are in no way compromised by the projection.

Similarly, we have used a somewhat standard color scheme: the rainbow in panel 1, and a warm/cool two-color palette for the difference plots. This again is quite customary.

We are somewhat puzzled by this remark as neither the projection nor the color scheme are in any way misleading or distorting the results.

That said, we have modified Fig. 5 and 6 to show the model-obs. differences in Panel (b), and the (smaller) model-model differences in Panels (c) and (d). We hope this makes the discussion clearer. See page 11, line 28, and captions to Figs. 5 and 6.

26. Figure 5: I would like also to see land temperatures. And differences among the simulations are far more interesting than differences with GPCP

Again, we are puzzled by the remark and unsure how (or why) to add land temperatures to a global plot of precipitation. No land or ocean cells have been masked on this plot.

The differences between the simulations are interesting, but cannot be over-interpreted, as in practice we would retune the models to maintain the same top-of-atmosphere net radiative flux. The focus of the paper is on the computational aspects and the coupling algorithm. We expect to visit this issue in greater detail in a future paper describing scientific model results from concurrent-radiation run where $\Delta t_{\rm rad} = \Delta t_{\rm atm}$, with appropriate tuning. See also response to (28) below.

27. P10l14: "Remarkably" similar. . . my rule of thumb is that people use the word remarkable when they don't know what of substance to remark.

Rephrased, see page 12, line 2.

28. p10l30: I would make the tuning point later, as it seems as though the authors are interpreting the differences as fundamental, rather then simply an illustration of compensating biases in a manner that is to be anticipated.

The tuning discussion comes at the tail-end of Section 4.1, which is the last discussion of physical results. We agree that the differences are not fundamental, and in the initial draft had indicated that they were "within the margins of the tuning process." We have expanded on that to make it clearer, as suggested, see page 13, line 26.

29. p11l5: SYPD has a unit, i.e., yr d≂1

SYPD is itself a unit, in fact it is introduced using the phrase "simulated years per day, or SYPD", see page 13, line 30. We believe it would be redundant to attach the unit again. Units are mandatory when the same quantity can be expressed in different units, e.g m and cm. That is not so in this case.

That said, we have added a discussion and definition of SYPD, see page 13, line 30.

30. Table 1: Maybe spell out the acronyms; for example does the introduction of CHSY really help anything? And if so why not PHSY, processor hours per simulated year. The computational cost of radiation appears small. If increasing the frequency of radiation nine-fold increases the computational cost by 50% this implies that the cost of radiation is about 5% of the total computational cost in the default configuration. Is this correct? If so this is rather small compared to some other models, suggesting that the proposed approach might be even more beneficial for other centers, or offer the possibility of more exact representations of radiative transfer. Here some clear numbers would be useful.

We do indeed believe the CHSY discussion is useful, as both time to solution (SYPD) *and* resource consumption (CHSY) are taken into account in configuring the parallel layout of a production model. See discussion in Balaji et al. (2016). We have also chosen to be consistent with that paper in naming the throughput measure CHSY instead of PHSY, which we agree would have also been a valid choice.

In regard the second point, the reviewer is correct, and we have made this point now in the text, see page 14, line 7.

31. p13l13: This paragraph is a bit ungrounded in the manuscript, which does not evaluate MPMD approaches. Certainly the GPU rewrite of COSMO has a factor of 3.6 speed up on a first implementation . . . so there is some room for efficiency gains through reprogramming, also the inexact hardware approaches (Dübben and Palmer) merit mention if this paragraph were to be retained and better grounded in the manuscript.

It's true that the background material is a bit ungrounded in *this* manuscript, but summarizes the findings of Balaji (2015), which does indeed survey MPMD methods and inexact computing approaches. We have rewritten the paragraph to reflect that these are findings from Balaji (2015). See page 15, line 18.

32. p13l20: Where does the order ten components come from. I think "probably not more than ten" would be more accurate, but in either case if this comes at the end it should also be better ground in the manuscript.

We have toned down the speculative statements here, see page 16, line 10.

33. p14l22: By data do the authors mean model output? Or the performance data, gleaned from the benchmarks? The use of "data" suggests the latter, but the former should also be addressed.

Fixed, see page 17, line 3

Response to `gmd-2016-114-RC2`:

34. page 2, around line 20: the discussion focuses on performance aspects only. Mention of the implications for energy/power use would also be useful here, as the focus is on extreme-scale systems.

    Fixed, see page 2, line 19

35. page 3, line 12: "how it is achieved without increasing data movement" should, I think, be changed to "and how it is achieved with minimal impact on data movement". This point is discussed further below [*].

    [*] The following point is my main concern with the paper as it stands. I believe the paper would benefit by being clearer on the use, and limitations, of shared memory threading to implement the concurrent execution of the radiation and the rest of the atmosphere model. It is clear that for a given multi-core processor there will be limits on the number of MPI processes per node and the number of threads each model may use within an MPI process without incurring potentially expensive data movement between caches. The example results given are based on two threads for each model. It would be good to make clear the rationale for this choice. Here are some thoughts on this and suggestions for possible changes which might help achieve this. Currently, the use of threads for executing the radiation in parallel with the rest of the atmosphere is described as not incurring any communication costs. While it is true there will be no MPI communication incurred between cores running the atmosphere and cores running the radiation, I believe there may well be some extra remote data accesses (i.e. cache misses) incurred between cores running the atmosphere and cores running radiation. The magnitude of this effect is, of course, architecture dependent and also depends on the number of threads used for each model and the mapping of the threads to cores. The results presented are for AMD Interlagos processors with two threads used for the atmosphere model and two threads used for the radiation model (within each MPI process). The Interlagos processor chip consists of eight 2-core modules. Two threads executing on the same module share an L2 cache. All 2 core modules share a large L3 cache. So, if one atmosphere and one radiation thread share a module, they can share data in the L2 cache (as well as the L3 cache). If two Atmosphere threads share a module and two radiation threads share a module, they will communicate through the L3 cache, which is more expensive in terms of cycles to access. If threads are on separate processor chips, there will be (even more expensive) data movement within the shared memory node.

    The cache behaviour in either of the above cases is likely to be different to that of a single thread running first the atmosphere and then the radiation. If more than two threads were used for each model, some sharing would have to take place via the L3 cache. Total thread numbers are clearly limited by the core count of a shared memory node. I would suggest to the authors that some clarification of these issues be made. For example: - in Figures 3 and 4, the images depicting MPI and OpenMP could be re-drawn to illustrate the relationship of threads within MPI in each case. In Figure 3, this would simply show multiple threads in an MPI task and multiple MPI tasks. In Figure 4, MPI tasks could be shown with both atmosphere and radiation threads or with ocean threads. For Figure 4, this might be something like: [AARR] [AARR]... [O] [O]... (where [] here represent MPI processes and letters represent threads and their models) - In Section 3 (perhaps?), a brief description of a multi-core processor (like the Interlagos) could be given along with the implications of thread-to-core mapping. This description would help to explain the benefit of using OpenMP to exploit

parallelism between the atmosphere and radiation models (i.e. no MPI communication) and pave the way for a discussion of the potential for sharing data between caches in the specific configurations presented in the results section.

The concurrency software design was formulated to ensure the radiation component was truly independent from the remaining atmosphere components with a single data synchronization point (copy) at the end of each time step. The data copying, itself implemented with OpenMP threading, can be completely removed via index flipping, but this might result in non-local accesses or extra cache invalidations - so a simple data copy is preferable. Additionally, each thread immediately calls a subroutine (either atmosphere or radiation) and is working with either explicitly blocked data structures or pseudo-blocked data created via copy-in variables. So not only are the atmospheric and radiation components completely independent, but within each component the data for each thread is isolated as well. If a radical paradigm shift in computing occurs, this forethought in the design allows us to quickly re-cast the radiation threads as separate MPI processes with a needed MPI communication synchronization point replacing the current data copy. See also (5) above.

36. page 5, Figure 1: The role of A_t should be depicted in the top figure (to be consistent with equations 1 and 2), I feel. Also, in many models, the atmosphere is executed on more processors than the ocean (because it scales better). Is this diagram consistent in this respect with the FMS model being described? Also, the bottom figure in Figure 1 implies that the ocean is executed on fewer processors in the concurrent set up. Is the intention to simply show a deployment utilising the same number of processors in total? If so, that should be made clear in the caption and text.

The relative balance of PE count between atmosphere and ocean depends on many factors, and within the FMS system, we have examples of both. However, in Figure 1 this is mostly schematic, indicating that some PEs may idle in serial processing.

We have adjusted the numbering of the component timesteps to be consistent with Eqs. 1-4.

37. page 9, line 2: "chosen to offer optimal load balance" could be extended to "chosen to offer optimal load balance and data sharing", for example. [I generally have a concern over the use of the work "optimal", which has a formal sense of "provably best". The word "good" might be better unless the load balance is provably optimal?]

Fixed, see page 10, line 1

38. page 11, line 9: The above arguments are tied up with the statement that "All runs use the optimal processor/thread layout for a given PE count". Some explanation about what this layout is and how it was chosen could be added.

Done, see page 13, line 33

39. page 1, line 4: I suggest changing "based on marginal increases in clock speed" to, for example, "based on, at best, marginal increases in clock speed" since it is likely clock speeds may decrease in future in some systems.

Done, see page 1, line 4

40. page 1, line 14: Define the acronym CCC here.

   Done, see page 1, line 16

41. page 1, line 15: is a little ambiguous about what is running in parallel ("and all other atmospheric physics components". I would suggest making it clear that there are only two concurrent components (i.e. not all "other atmospheric components" are executed in parallel with each other!

   Done, see page 1, line 18

42. page 2, line 3: perhaps provide a reference to the IPPC assessments.

   See page 2, line 15.

43. Section 2: page 4, line 4: needs a closing bracket after "example".

   Done, see page 4, line 33

44. page 8, line 24: "Individual" should be "individual".

   Done, see page 9, line 32

45. page 9, Figure 4: In this figure, the Land and Ice models are shown as executing concurrently but this is not mentioned in the text. This should be explained (or made consistent with Figure 3).

   Fixed. It should indeed have been consistent with Fig. 3. See new Fig. 4.

46. page 10, line 9: "that that" should be "that".

   Done, see page 11, line 21

47. page 10, line 11: This sentence would benefit from having a reference added.

   Done, see page 11, line 23

48. page 10, line 13: Expand the acronym GPCP here as a definition.

   Done, see page 11, line 26

49. page 10, lines 15-17: The point here is, I think, that this result is counter intuitive. If that is correct, it would be worth stating.

   As noted above in (25) and (28), the differences between the runs are small and within the bounds of the tuning process. Differences should not be over-interpreted as they are likely to vanish upon tuning. See page 13, line 26 also.

50. page 11, line 3: "less expensive as..." should be "less expensive as the...".

   Done, see page 13, line 25

51. page 12, line 4: the figures for processor count and SYPD given in this line are rounded versions of those in Table 1. Those in the previous sentence are not rounded. Please use the precise figures for consistency.

Done, see page 15, line 4

52. page 13, line 18: It would be worth giving the definition of CCC again here to remind the reader.

5 Done, see page 16, line 2

Response to Topical Editor's comments:

53. My main concern relates to the issue #35 (referee 2) which is also linked to issue #5 (referee 1). In fact, I do not really understand your replies to these two important issues and I understand that you did not make any modifications in the document to address them. Thank you in advance for a revised version of the document taking into account these issues.

Section 3 has been largely rewritten in a manner which I hope will address your concerns, as well as those raised in #5 and #35.

54. #6: IPCC, AR4, GFDL, CM3, NOAA, PE, GFDL, CPU are still not defined

Fixed, see page 1, line 16; page 2, line 13; page 2, line 15; page 11, line 5; page 1, line 1; page 9, line 23; page 16, line 22.

55. #25 - #26. :

(a) I think the reviewer was not asking to add land temperatures to Fig 5 and 6 but just add an extra plot similar to Fig.5 and 6 for land temperatures. Is that possible?

With regards to the request to show the response of surface temperature to the radiative time step and the serial vs. concurrent coupling, we prefer not to do this since these integrations prescribe the ocean surface temperature, which prevents land temperatures from changing much as well. The changes in top of atmosphere radiative fluxes in an AMIP model that we are showing are more relevant to the temperature changes in a coupled model than are the temperature changes in an AMIP (prescribes sea surface temperature) model themselves.

(b) Fig. 6 captions do not read OK to me now. Please add "(a)." after "CERES EBAF v2.8 climatological top-of-atmosphere radiation budget" and change ", and (b) Model ..." for "Panel (b) shows model ..."

Fixed, see new caption to Figure 6

56. #23, p.9, l.23: I am not sure I understand your "PE/PElist" concept, as you are mixing hardware (core) and software (communicator) concepts. If a PElist is equivalent to the communicator concept in MPI, you refer here to the software concept that I would call a list of MPI tasks. But then in Table 1 legend, you write "NPES is the total PE count (MPI*OMP)" so it looks like there is no equivalence between a PE and an MPI task. If two processes share one core (multitasking), would you have two PEs (two processes) or only one PE (one core) in your PElist? Can you clarify?

You are correct: a PE is a hardware unit, and a PElist is an MPI communicator. We have clarified the text, see page 9, line 26.

57. #36 I think the new numbering of the component timesteps in the top panel of Fig. 1 is wrong and the previous was right. Eq. (1) states that $A^{t+1} = f(A^t, O^t)$, and $O^{t+1} = f(O^t, A^{t+1})$. The top panel of Fig. 1 would imply $A^{t+1} = f(A^t, O^{t+1})$ and $O^{t+1} = f(A^t, O^t)$.

I think I have got it right this time, see new Figure 1!

58. #49.: I understand your argument about differences probably vanishing with tuning, but as referee 2, I still think that the results that are counter intuitive and this should be stated. Also the sentence "Although this difference is small we feel it is robust and likely related to a sensitivity to the radiative time step discussed below." seems contradictory to me. If they are robust, they would not vanish with tuning?

We have rewritten the discussion on tuning, which we hope is clearer, see page 11, line 25; page 12, line 2; page 13, line 2; page 13, line 21. The main point is that the magnitude is within the bounds of tuning, so we perhaps not pay too much attention to features within those bounds.

59. Fig.3 and associated text: even if the captions help understand the FMS architecture, I still agree with the referees that some aspects are misleading.

   (a) First, I would remove the green openMP box and the dark blue MPI box on the figure (as the meaning of the colours is now clearly explained in the text. Second, for coherency, I would replace "Atmos Down" by "atmosphere-down" and "Atmos Up" by "atmosphere-up" in the figure, or put "(Atmos Down)" after "atmosphere-down" and "(Atmos Up)" after "atmosphere-up" when they appear in the text.

   Fixed: see page 8, line 5 and new Figures 3 and 4.

   (b) In the text, you write "The implicit coupling requires ... (land and ocean) ..." but I understand that FMS implements the implicit coupling only over the land and not over the ocean. Can you clarify this?

   Actually, it is implicit also over the ocean surface, which includes sea ice. The explicit coupling is with the 3D ocean, see Balaji et al. (2006) for details.

60. p.2, l.13 please change "... assessments, and their complexity (the number of feedbacks and phenomena simulated), exhibits ..." for "... assessments and their complexity (the number of feedbacks and phenomena simulated) exhibit ..."

   Fixed, see page 2, line 13

61. p.2, l.14 please change "Figure 1.2[1] Figure 1.4[2]" for "Figure 1.2[1] and Figure 1.4[2]"

   Fixed, see page 2, line 15

62. p.3, l.2: please change "operatorss" for "operators"

   Fixed, see page 3, line 4

63. p.3, l.2: why do you qualify "operators and operands" as "new"?

   They are "new" because new instructions keep arriving in the processor during each context switch. I have changed it to "fresh", which I hope is clearer: see page 3, line 4

64. p.3, l.3: please change "... and locality and reuse hard to achieve." for "... and locality and reuse are hard to achieve."

   Fixed, see page 3, line 5.

65. p.3, l.23: please move "(temporal subsampling)" right after "coarser timestep than the rest of the atmosphere"

Fixed, see page 3, line 25

66. p.3, l.25: I do not understand the "as well as the temporal". Do you mean "as well as in the temporal domain"?

Fixed, see page 3, line 27

5    67. p.4, l.7: please change "... AMIP (The Atmospheric ..." for "... AMIP (the Atmospheric ..."

Fixed, see page 4, line 10

68. In Table 1 legend, please replace "MPI/OMP" by "MPI*OMP"

Fixed, see legend to Table 1.