# Peer review of "Coarse-grained component concurrency in Earth System modeling: parallelizing atmospheric radiative transfer in the GFDL AM3 model using the Flexible Modeling System coupling framework"

_Geoscientific Model Development, 2016_

## Short Comment (SC1) · 6 Jun 2016

Dear authors,

In my role as Executive editor of GMD, I would like to bring to your attention our Editorial version 1.1:

http://www.geosci-model-dev.net/8/3487/2015/gmd-8-3487-2015.html

This highlights some requirements of papers published in GMD, which is also available on the GMD website in the 'Manuscript Types' section:

http://www.geoscientific-model-development.net/submission/manuscript_types.html

In particular, please note that for your paper, the following requirements have not been met in the Discussions paper:

- "The main paper must give the model name and version number (or other unique identifier) in the title."

- "If the model development relates to a single model then the model name and the version number must be included in the title of the paper. If the main intention of an article is to make a general (i.e. model independent) statement about the usefulness of a new development, but the usefulness is shown with the help of one specific model, the model name and version number must be stated in the title. The title could have a form such as, "Title outlining amazing generic advance: a case study with Model XXX (version Y)"."

Please add the name if the GFDL model and its version number to the title when you revise the article for submission to GMD.

Yours,

Astrid Kerkweg

---

## Referee Comment (RC1) · Anonymous Referee #1 · 28 Jun 2016

General Comments:

This is a well composed manuscript written in a lively style to describe experiments in finer grained component concurrency for at atmospheric model. The novel aspect is the use of component concurrency to improve scalability in an atmospheric code. In general the manuscript is appropriate for GMD, but revisions would strengthen the presentation.

The suggested revisions are substantial, but of a nature that the editor can readily adjudicate.

Specific Comments:

1. I enjoyed the more lyrical style, but in places the style came across as glib, particularly when precision was sacrificed for poetry. Grounding the manuscript better in data and substance would strengthen the exposition greatly.

2. The figures are substandard and not sufficiently quantitative. In particular differences between different configurations of the same simulation are preferred over differences to the observations, as the former, not the latter, is the point of the manuscript. Also attention to map projections and color scales is required.

3. The frequent reference to Pauluis and Emanuel was insufficiently discriminating and in places misleading. The reference gives the impression that infrequent coupling to radiation is a substantial source of bias and also model instability. This is not what the manuscript is about. Moreover the Pauluis Emanuel study, while noteworthy, has not been shown to generalize. It might, but the literature is not there. In the grand scheme of things, the trade-off in accuracy and stability of calling radiation more frequently, versus simulating at higher resolution, is not well understood.

4. Some of the hard-core computational issues are insufficiently addressed. In particular, component concurrency probably will affect the hard scaling floor, and the trade-off between communication vs processing in codes like ESMs with low arithmetic intensity. I guess there are also trade-offs that arise because of the need for the concurrency to be in a shared memory implementation. The manuscript would be strengthened if these issues were discussed in a more thorough manner.

5. There are too many acronyms, and some seem indiscriminately chosen.

6. The manuscript does not distinguish between AGCMs and ESMs, frequently discussing the results in terms of ESMs but then presenting results for the atmospheric GCM alone. Do the results generalize to problems that already have much more concurrency? I guess so but this is a separate point and the manuscript should discriminate between what was done and what is inferred based on what was done.

Details:

p1l2: Acronyms

p1l24: Is that true, or did the technology also drive things. it makes it sound like everything was possible and we just chose something, rather than necessity driving development.

p2l5: I am not sure what figure the authors are trying to say here. Actually resolution has not kept pace with computing as far as I can tell, and the reference to the figure does not make sense.

p2l7: Would help to explain to the reader the phrase "arithmetic and logic" is a memory fetch logic?

p2l11: "The state of play of climate computing in the face of these challenges" this phrase comes across as a bit of a throw away. Did the Balaji (2015) paper make a point that is important for the present discussion? If so what was it.

p2l20: "and there is constant churn of operations " missing an article. . . lyrical, but it gives the idea that the computer is kept busy computing rather than moving information around. Most codes are memory bandwidth limited . . . which you get to shortly. But this intro sentence did not prepare me well.

p2l30: Here: "Of the many factors of 10 increase in performance needed to get to the promised land of "exascale computing", we believe at least one can come from component organization. " I would prefer precision over poetry.

p3l5: Weather centers also run spatially coarse-grained radiation. Also some have proposed a form of coarse graining in the spectral domain, i.e., Monte-Carlo Spectral Integration.

p4l9: A reference is needed here. My intuition suggests that such high processor counts have only been applied to more traditional Atmosphere, or Ocean or Atmosphere Ocean Problems, but not simulations of the carbon cycle, i.e., the use of ESM here is misleading as earlier it implied biology.

[Figure]

   The reviewer is correct: most of the very high end PE counts are for specialized problems, not full ESMs in climate mode. We have clarified this, and added some references: see page 5, line 9

p6l15: Subscript abbreviations are usually written in roman font.

   Fixed, here and everywhere else in the text: see page 7, line 15.

Figure 3: I spent some time on this and I am not sure I understood it. The vertical dimension denotes sequence, first to last from top to bottom. The boxes indicated either the legend or a component process? The thickness of the box denotes? A figure should be illustrative, not a riddle. Also a bit more structure might help me understand what atmosphere up is. The meaning of some colors seems to be specific, others decorative?

p7 l2: 'interact strongly with atmospheric chemical species and clouds' this could exclude water vapor, why not say, couple strongly to composition.

p7 l4: By this definition the ocean does not have tracers. Aren't tracers really just scalar quantities that are transported with the flow subject to source and sink processes.

p7 l6: Seconds and hours (line 9) are units, and can be abbreviated.

p7 l9: This makes it sound worse than it may be, some codes rescale the radiative heating rates at each timestep by the insolation, or the surface temperature, in a sense linearizing about the state defined every 3 hrs. Something you mention on the next page, but it comes late.

p7 l19: Spell out PE . . . Processing Elements? A socket?

p7 l14: 'Architected'? Okay, it can be used as a verb; but I think designed, or constructed would be better.

Figure 5: This needs redrafting, first for the colour scale (no rainbows); second to show the common color scales in those panels where it is appropriate. The highly distorted projection should either be motivated or replaced.

Figure 5: I would like also to see land temperatures. And differences among the simulations are far more interesting than differences with GPCP

P10l14: 'Remarkably' similar. . . my rule of thumb is that people use the word remarkable when they don't know what of substance to remark.

p10l30: I would make the tuning point later, as it seems as though the authors are interpreting the differences as fundamental, rather then simply an illustration of compensating biases in a manner that is to be anticipated.

p11l5: SYPD has a unit, i.e., yr dˆ-1

Table 1: Maybe spell out the acronyms; for example does the introduction of CHSY really help anything? And if so why not PHSY, processor hours per simulated year.

The computational cost of radiation appears small. If increasing the frequency of radiation nine-fold increases the computational cost by 50% this implies that the cost of radiation is about 5% of the total computational cost in the default configuration. Is this correct? If so this is rather small compared to some other models, suggesting that the proposed approach might be even more beneficial for other centers, or offer the possibility of more exact representations of radiative transfer. Here some clear numbers would be useful.

p13l13: This paragraph is a bit ungrounded in the manuscript, which does not evaluate MPMD approaches. Certainly the GPU rewrite of COSMO has a factor of 3.6 speed up on a first implementation . . . so there is some room for efficiency gains through reprogramming, also the inexact hardware approaches (Dübben and Palmer) merit mention if this paragraph were to be retained and better grounded in the manuscript.

p13l20: Where does the order ten components come from. I think "probably not more than ten" would be more accurate, but in either case if this comes at the end it should also be better ground in the manuscript.

p14l22: By data do the authors mean model output? Or the performance data, gleaned from the benchmarks? The use of "data" suggests the latter, but the former should also be addressed.

---

## Referee Comment (RC2) · Anonymous Referee #2 · 1 Jul 2016

Coarse-grained component concurrency in Earth System modeling V. Balaji, R. Benson, B. Wyman, and I. Held

General Comments

The paper describes an approach to exploit a previously unexploited level of concurrency from multi-component, coupled Earth System Models (ESMs). The work is motivated by a discussion of the trends in the development of extreme-scale high performance computing where future performance gains require the exploitation of vastly higher amounts of concurrency than previously required.

The motivation is very well argued with respect to the need for new techniques to exploit more concurrency in such models and from the algorithmic perspective. For example, in the specific case of attempting to execute the radiation model concurrently

with the rest of the atmosphere model, the resulting concurrent algorithm is known to be theoretically unstable but, nevertheless, the implementation is found to be stable in practice due to other physical processes.

The implementation of the new concurrent algorithm exploits the shared memory nature of modern multi-core CPUs in order to minimise the impact on data movement between the component models. This is a crucial aspect, since a traditional approach to exploiting model concurrency - running two models on separate sets of processes and exchanging data through a message passing-based model coupler - would lead to excessive data movement between processor cores that would severely impact performance.

The paper successfully demonstrates that there is much more concurrency in ESMs than has previously been exploited and that there are techniques that can exploit this concurrency effectively on the emerging extreme-scale HPC systems. The new algorithm allows the (expensive) radiation model to run more frequently (i.e with a smaller timestep, hence coupling more frequently) while maintaining the overall simulation rate while using more processor cores. This is a very encouraging story on the road to exascale for Earth System Modelling.

Specific Comments

page 2, around line 20: the discussion focuses on performance aspects only. Mention of the implications for energy/power use would also be useful here, as the focus is on extreme-scale systems.

page 3, line 12: "how it is achieved without increasing data movement" should, I think, be changed to "and how it is achieved with minimal impact on data movement". This point is discussed further below [*].

[revised manuscript text omitted]

- page 9, line 2: "chosen to offer optimal load balance" could be extended to "chosen to offer optimal load balance and data sharing", for example. [I generally have a concern over the use of the work "optimal", which has a formal sense of "provably best". The word "good" might be better unless the load balance is provably optimal?]

- page 11, line 9: The above arguments are tied up with the statement that "All runs use the optimal processor/thread layout for a given PE count". Some explanation about what this layout is and how it was chosen could be added.

Technical Corrections

page 1, line 4: I suggest changing "based on marginal increases in clock speed" to, for example, "based on, at best, marginal increases in clock speed" since it is likely clock speeds may decrease in future in some systems.

page 1, line 14: Define the acronym CCC here.

page 1, line 15: is a little ambiguous about what is running in parallel ("and all other atmospheric physics components". I would suggest making it clear that there are only two concurrent components (i.e. not all "other atmospheric components" are executed in parallel with each other!

page 2, line 3: perhaps provide a reference to the IPPC assessments.

Section 2:

page 4, line 4: needs a closing bracket after "example".

page 8, line 24: "Individual" should be "individual".

page 9, Figure 4: In this figure, the Land and Ice models are shown as executing concurrently but this is not mentioned in the text. This should be explained (or made

consistent with Figure 3).

page 10, line 9: "that that" should be "that".

page 10, line 11: This sentence would benefit from having a reference added.

page 10, line 13: Expand the acronym GPCP here as a definition.

page 10, lines 15-17: The point here is, I think, that this result is counter intuitive. If that is correct, it would be worth stating.

page 11, line 3: "less expensive as..." should be "less expensive as the...".

page 12, line 4: the figures for processor count and SYPD given in this line are rounded versions of those in Table 1. Those in the previous sentence are not rounded. Please use the precise figures for consistency.

page 13, line 18: It would be worth giving the definition of CCC again here to remind the reader.

―――――――――――――――

---

## Author Comment (AC1) · 9 Aug 2016

Response to `gmd-2016-114-SC1`:

1. The main paper must give the model name and version number (or other unique identifier) in the title. If the model
   development relates to a single model then the model name and the version number must be included in the title of the
   paper. If the main intention of an article is to make a general (i.e. model independent) statement about the usefulness of a
5  new development, but the usefulness is shown with the help of one specific model, the model name and version number
   must be stated in the title. The title could have a form such as, "Title outlining amazing generic advance: a case study
   with Model XXX (version Y)".

   In response, we have modified the title as follows: "Coarse-grained component concurrency in Earth System modeling:
   Parallelizing atmospheric radiative transfer in the GFDL AM3 model using the Flexible Modeling System coupling
10 framework".

---

## Editor Decision (ED1)

Dear author,

Thank you for your revised manuscript and answers to the referee's questions and comments. They are quite complete but I consider that some revisions are still needed.

My main concern relates to the issue #35 (referee 2) which is also linked to issue #5 (referee 1). In fact, I do not really understand your replies to these two important issues and I understand that you did not make any modifications in the document to address them. Thank you in advance for a revised version of the document taking into account these issues.

I also have the following additional remarks:

- #6: IPCC, AR4, GFDL, CM3, NOAA, PE, GFDL, CPU are still not defined
- #25 - #26. :
  - I think the reviewer was not asking to add land temperatures to  Fig 5 and 6 but just add an extra plot similar to Fig.5 and 6 for land temperatures. Is that possible?
  - Fig. 6 captions do not read OK to me now. Please add "(a)." after  "CERES EBAF v2.8 climatological top-of-atmosphere radiation budget " and change ", and (b) Model ..." for "Panel (b) shows model ..."
- #23, p.9, l.23: I am not sure I understand your "PE/PElist" concept, as you are mixing hardware (core) and software (communicator) concepts. If a PElist is equivalent to the communicator concept in MPI, you refer here to the software concept that I would call a list of  MPI tasks. But then in Table 1 legend, you write "NPES is the total PE count (MPI*OMP)" so it looks like there is no equivalence between a PE and an MPI task. If two processes share one core (multitasking), would you have two PEs (two processes) or only one PE (one core) in your PElist? Can you clarify?
- #36 I think the new numbering of the component timesteps in the top panel of Fig. 1 is wrong and the previous was right. Eq. (1) states that $A^{t+1}=f(A^t,O^t)$, and $O^{t+1}=f(O^t, A^{t+1})$. The top panel of Fig. 1 would imply $A^{t+1}=f(A^t,O^{t+1})$ and $O^{t+1}=f(A^t,O^t)$.
- #49.: I understand your argument about differences probably vanishing with tuning, but as referee 2, I still think that the results that are counter intuitive and this should be stated. Also the sentence "Although this difference is small we feel it is robust and likely related to a sensitivity to the radiative time step discussed below. " seems contradictory to me. If they are robust, they would not vanish with tuning?
- Fig.3 and associated text: even if the captions help understand the FMS architecture, I still agree with the referees that some aspects are misleading.
  - First, I would remove the green openMP box and the dark blue MPI box on the figure (as the meaning of the colours is now clearly explained in the text.
  - Second, for coherency, I would replace "Atmos Down" by "atmosphere-down" and "Atmos Up" by "atmosphere-up" in the figure, or put "(Atmos Down)" after "atmosphere-down" and "(Atmos Up)" after "atmosphere-up" when they appear in the text.
  - In the text, you write "The implicit coupling requires … (land and ocean) ..." but I understand that FMS implements the implicit coupling only over the land and not over the ocean. Can you clarify this?

Finally, I have the following additional minor comments:

- p.2, l.13 please change "... assessments, and their complexity (the number of feedbacks and phenomena simulated), exhibits ..." for "... assessments and their complexity (the number of feedbacks and phenomena simulated) exhibit ..."

- p.2, l.14 please change "Figure 1.2[1] Figure 1.4[2] " for "Figure 1.2[1] and Figure 1.4[2] "
- p.3, l.2: please change "operatorss" for "operators"
- p.3, l.2: why do you qualify "operators and operands" as "new"?
- p.3, l.3: please change "... and locality and reuse hard to achieve." for "... and locality and reuse are hard to achieve."
- p.3, l.23: please move "(temporal subsampling)" right after "coarser timestep than the rest of the atmosphere"
- p.3, l.25: I do not understand the "as well as the temporal". Do you mean "as well as in the temporal domain"?
- p.4, l.7: please change "... AMIP (The Atmospheric ..." for "... AMIP (the Atmospheric ..."
- In Table 1 legend, please replace "MPI/OMP" by "MPI*OMP"